



# Stochastic properties of coastal flooding events – Part 1: CNN-based semantic segmentation for water detection

Byungho Kang[2,1], Rusty A. Feagin[3,1,*], Thomas Huff[3,*], and Orencio Durán Vinent[1]

[1]Department of Ocean Engineering, Texas A&M University, College Station, TX, USA
[2]Department of Civil and Environmental Engineering, University of Houston, Houston, TX, USA
[3]Department of Ecology and Conservation Biology, Texas A&M University, College Station, TX, USA
[*]These authors contributed equally to this work.

**Correspondence:** Orencio Durán Vinent (oduranvinent@tamu.edu)

**Abstract.** The frequency and intensity of coastal flooding is expected to accelerate in low-elevation coastal areas due to sea level rise. Coastal flooding due to wave runup affects coastal ecosystems and infrastructure, however it can be difficult to monitor in remote and vulnerable areas. Here we use a camera-based system to monitor wave runup as part of the after-storm recovery of an eroded beach on the Texas coast. We analyze high-temporal resolution images of the beach using Convolutional

Neural Network (CNN)-based semantic segmentation to study the stochastic properties of runup-driven flooding events. In the first part of this work, we focus on the application of semantic segmentation to identify water and runup events. We train and validate a CNN with over 500 manually classified images, and introduce a post-processing method to reduce false positives. We find that the accuracy of CNN predictions of water pixels is around 90% and strongly depend on the number and diversity of images used for training.

## 10  1   Introduction

Coastal flooding can cause significant damage to coastal ecosystems and infrastructure and greatly affect the functioning of coastal communities. This flooding occurs when extreme water levels—due to a combination of high tide, storm surge and wave runup—exceed a natural or artificial barrier. The frequency and severity of coastal flooding is expected to increase with the acceleration of sea level rise (Nicholls et al., 2011; Vitousek et al., 2017). In order to respond to and minimize the damage

from coastal flooding, it is crucial to determine the frequency and intensity of flooding events at different locations and identify the physical factors behind them (Hallegatte et al., 2013; Moore and Obradovich, 2020).

Over spatial scales on the order of kilometers or larger, and on time scales on the order of minutes or larger, water levels are usually estimated from tidal gauges, while existing predicting tools can also account for wind-induced water levels (Kang et al., in review). Although these results can be interpolated fairly well to cover locations between gauges, they still fail to

capture the contribution of local wave runup. Wave runup, most typically measured by $R_{2\%}$, can greatly exceed the average predicted water level by other methods or sensors, resulting in the overtopping of coastal dunes and seawalls. This is because runup is a function of wave height relative to wave length (Battjes, 1974) and depends on wave interaction with the bathymetry and topography (Strauss et al., 2012; Vitousek et al., 2017). In spite of several empirical formulas to estimate wave runup from



offshore wave data (e.g., Stockdon et al., 2006, 2014), accurately predicting wave runup is difficult because wave interaction
is spatially localized. Furthermore, wave height and length are individualized measures that longer temporal scale gauges are
not designed to detect.

Fortunately, the excursion extent of the water level at these localized spatial and temporal scales can be generally represented
by the existence of the wet/dry line along a sandy beach. Therefore, a camera-based system designed to resolve wave runup
events could be a powerful tool to monitor coastal flooding and also gather high-quality data to improve empirical formulas
and increase our understanding of runup's stochastic properties.

Recent developments in coastal camera systems can help identify coastal flooding along a coastline. Indeed, much data is
available online that could potentially be mined to help improve coastal flooding predictions if we had automated methods to
classify the wet/dry line in these images. For example, Vousdoukas et al. (2011) applied a classical machine-learning model,
with a three-layer Artificial Neural Network (ANN), to determine the pixel intensity threshold and estimate the elevations of
shoreline contours. Likewise, Alvarez-Ellacuria et al. (2011) applied ANN to a time exposure image to determine the shoreline.
Using structured Support Vector Machine (SVM), Hoonhout et al. (2015) estimated beach width and the location of the water
line based on semantic classification of mid-range coastal imagery, and proved the robustness of such technique for the long-
term analysis of the coastal imagery.

An important limitation of the aforementioned computer vision techniques is that they require calibration or feature-extraction
pre-processing at the initial stage. Therefore, in-depth knowledge was essential for each method, which has prevented their
widespread use. This limitation can be overcome using "deep learning" algorithms such as Convolutional Neural Networks
(CNN).

The "deep learning" movement started in the mid-2000s when Hinton et al. (2006) rekindled the use of the neural network
in machine learning by showing the networks with many hidden layers could also be trained as well. Following this work, the
introduction of the Rectified Linear Unit (RELU) for multi-layer back-propagation has led to the widespread use of CNNs for
image recognition, which inherently has deep architecture (Nair and Hinton, 2010; LeCun et al., 1989). Breakthroughs of Deep
Convolutional Neural Networks in image classification have been transferred to pixel-level semantic segmentation (Chen et al.,
2016), since a fully convolutional network based on decoder structure outperforms other classical machine learning models in
terms of pixel accuracy (Long et al., 2015).

Image segmentation based on CNN has been applied to various video-based coastal studies. Buscombe and Ritchie (2018)
introduced a hybrid model that combines fully connected conditional random field (CRF) and CNN platform to analyze large-
scale coastal imagery. Valentini and Balouin (2020) used the same method with the base of SLIC super-pixels instead of fixed
tiles to detect and monitor *Sargassum* algae for an early warning system. The approach was convenient and accurate, as it relied
on a predefined dataset for the classification, which does not require exhaustive manual annotations. However, this segmentation
process was based on the classification of tiles–a bundle of pixels–rather than actual "pixels". Thus, the minimal resolution
was often too low to classify features smaller than the size of tiles and superpixels. Furthermore, Sáez et al. (2021) used an
U-net architecture for detecting wave-breaking nearshore while other studies had tried to validate semantic segmentation by
comparing with other measurement methods. For example, by comparing the results with gauges in a physical model test, den



Bieman et al. (2020) showed that image segmentation by SegNet can reasonably predict surface elevation, run-up, and bed
level from video images.

In terms of flooding management, studies conducted by Muhadi et al. (2021) and Vandaele et al. (2021) have shown the
reliability of image segmentation for fluvial water level estimation. In those studies, the correlation between estimated water
level and the water level estimation from Lidar data and river gauge measurement was higher than 0.9, which signals potential
use for coastal flooding analysis.

In this work, we explore the use of CNN-based semantic segmentation to automatically detect water on beach imagery and
to identify and quantify coastal flooding events driven by wave runup. In addition to a brief introduction to CNN-based image
segmentation, we discuss the specific methodology used for this study and present a simple but powerful post-processing
method for refining the accuracy of semantic segmentation. We then investigate the performance of the method as function of
morphological diversity and the number of images in the training set.

## 2   Methodology

In order to test CNN methods, we collected field data at a heavily-eroded location after Hurricane Harvey struck the Texas
Coast in 2017 and explored the imagery using CNN. We provide a basic overview of CNN and the Deeplab architecture used
in our analysis and discuss the collection and annotation of the images used for training and validating the CNN.

### 2.1   Semantic segmentation using CNN

For the CNN architecture, we selected the Deeplab v3+ model based on Resnet-18, which adds some unique features to
semantic segmentation such as the Atrous or Dilated convolution. Independently introduced by Chen et al. (2016) and Yu and
Koltun (2015), embedding the holes to the convolution filters has helped to circumvent the resolution loss from downsampling,
by enlarging the field of view with a lower computational cost. Furthermore, batch normalization layers have been included in
each of the parallel modules in the Atrous Spatial Pyramid Pooling (ASPP). Normalization over a mini-batch minimizes the
transition in the distribution of internal network nodes (covariate shift) and assures the distribution of non-linear inputs remain
more robust (Ioffe and Szegedy, 2015). This helped to fix the vanishing gradient problem, and hence accelerated the training of
the deep neural network. Concatenated results from the parallel modules and image-level features were passed through $1 \times 1$
convolution, and provide more accurate results (Chen et al., 2017).

Deeplab v3+ also has a unique decoder module to refine the details of object boundaries which extends the previous versions.
Instead of naive bilinear upsampling the encoder features by a factor of 16, the encoder features are first bilinear upsampled
by a factor of 4 and concatenated with the result of $1 \times 1$ convolution of low-level features (Chen et al., 2018). The $3 \times 3$
convolutions are then applied for the refinement, which is followed by another upsampling by a factor of 4.

In this study, we conduct regular convolution instead of depthwise separable convolution typically used for Deeplab v3+
structure, as the Resnet-18 has relatively shallow layers (18 layers) and the computation burden is not heavy, such that we do
not have to reduce the number of weights in the cost of accuracy.



## 2.2 Data acquisition and classification

We installed three solar-powered stationary GoPro cameras, with different field of view, near Cedar Lakes in Texas (N 28.819°,W 95.519°) to monitor beach recovery after hurricane Harvey in 2017 (Fig.1). This site is subject to frequent wave runup events due to its low-lying, yet complex, bathymetric-topographic profile. We preset each camera to capture pictures every 5 minutes during a 6 A.M–6 P.M. observation period and turn off automatically during the nighttime. From November 2017 to May 2018, we captured more than 51,000 images.

We chose 584 random pictures with different scenery, weather and light conditions (Fig. 2) and manually labelled the regions on each picture according to three classes: "water", "sky", and "background", where the "sky" class was added to reduce false positives of "water" pixels (Fig. 3). We refer to the classes as labels when describing the pixel in an image.

Every pixel was hand-annotated for each class to improve accuracy. We opted not to utilize the Simple Linear Iterative Clustering (SLIC) superpixel method, a commonly employed computer vision technique for labeling, as it often resulted in clusters of pixels containing more than two classes. We then divided the 584 annotated pictures into a training set with 493 images (85%) and a validation set with 91 images (15%).

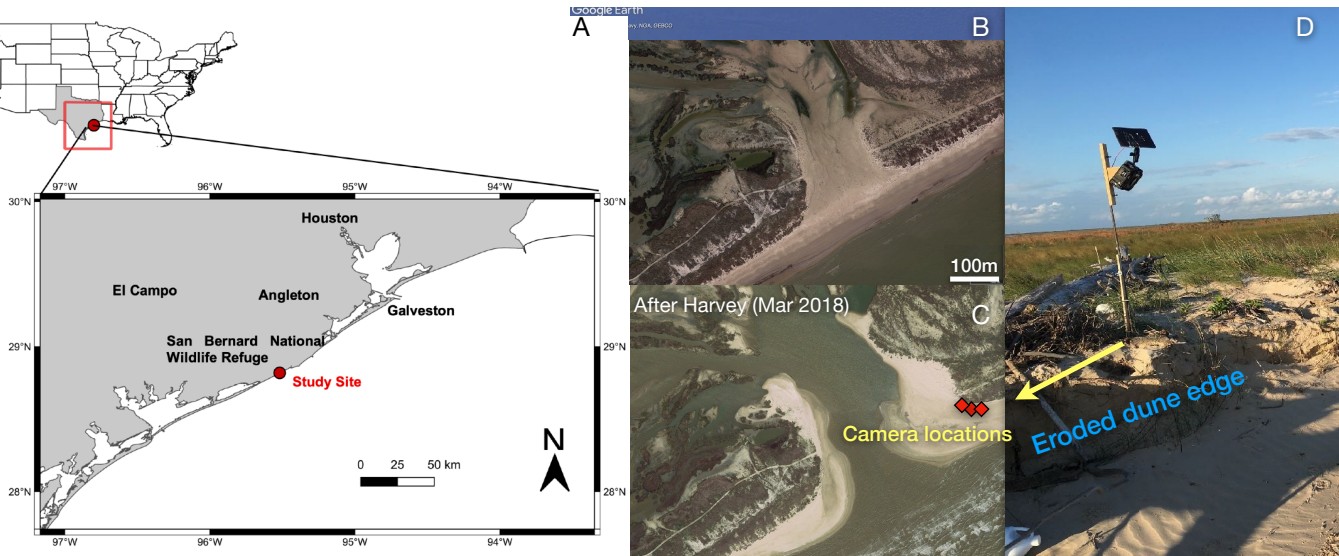

**Figure 1.** Location of field observations (A). Three solar-powered cameras (C-D) were installed in Cedar Lakes, Texas, a site breached during hurricane Harvey that experience frequent wave runup flooding.

## 2.3 Training Protocol

We used transfer learning (Bengio, 2012), which can expand the use of Deep Learning for the limited training data-set. We transferred the parameters of Resnet-18 that had been pre-trained using the subset of the ImageNet database for ImageNet





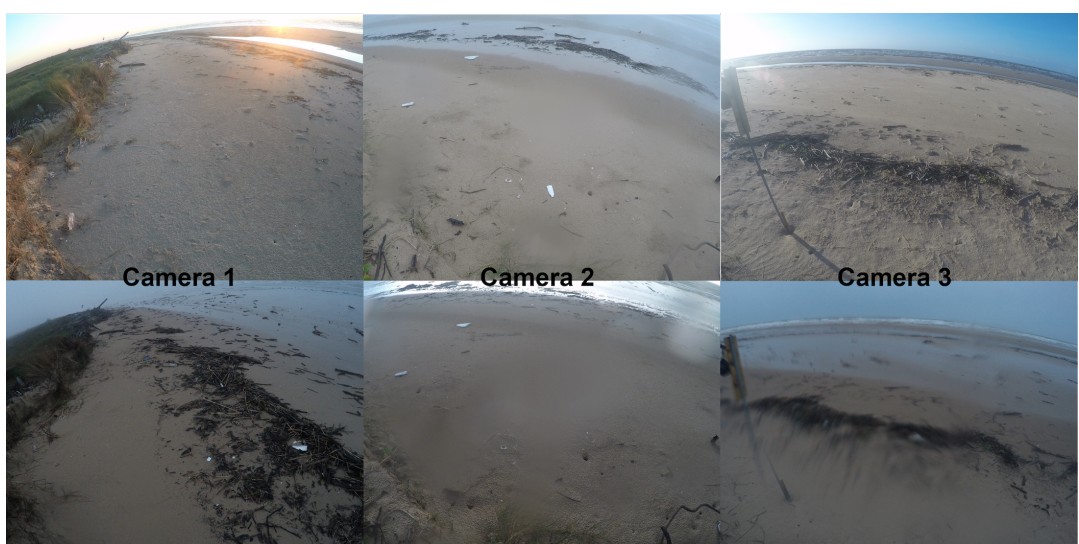

**Figure 2.** Contrasting images captured from our three cameras depicting both flooding and normal conditions.

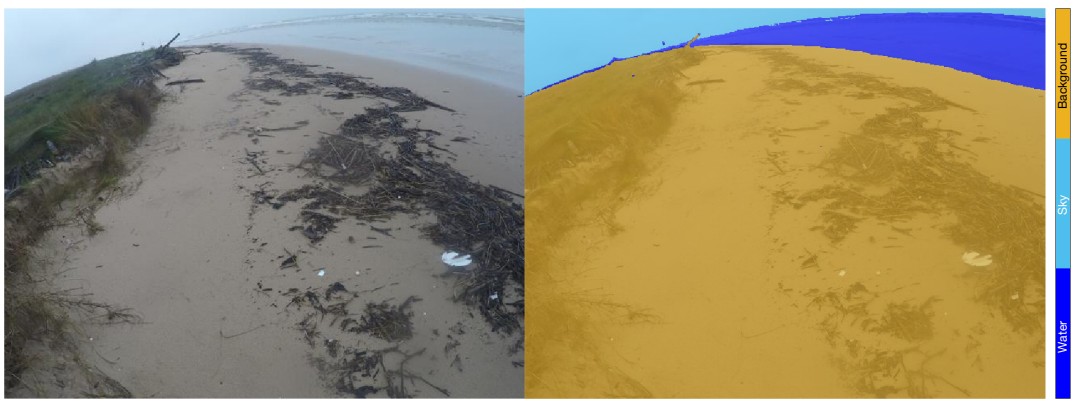

**Figure 3.** A example of a picture taken during flooding conditions (left) with manual labels superimposed (right).

Large-Scale Visual Recognition Challenge (ILSVRC) (Russakovsky et al., 2015). After this, we re-train the network with our training images using Matlab Deeplearning Toolbox as a module.

Both the training images and the manual annotations were compressed from $1920 \times 2560$ to $480 \times 640$. For every iteration during the training, we flipped each image horizontally with 50% probability to avoid over-fitting. The mini-batch size was 4. To deal with local minimum problem, images were shuffled randomly for every epoch.

We choose Stochastic Gradient Descent (SGD) with momentum for the optimization algorithm (Wilson et al., 2017), and set the momentum as 0.9. We also replaced the standard classification layer with the classification layer that use the weighted cross-entropy loss. This was done in order to offset the imbalanced classes induced by the backgrounds which covers more than 80% in almost every images (Eigen and Fergus, 2014). Additionally, we used the step decay learning rate for the training,



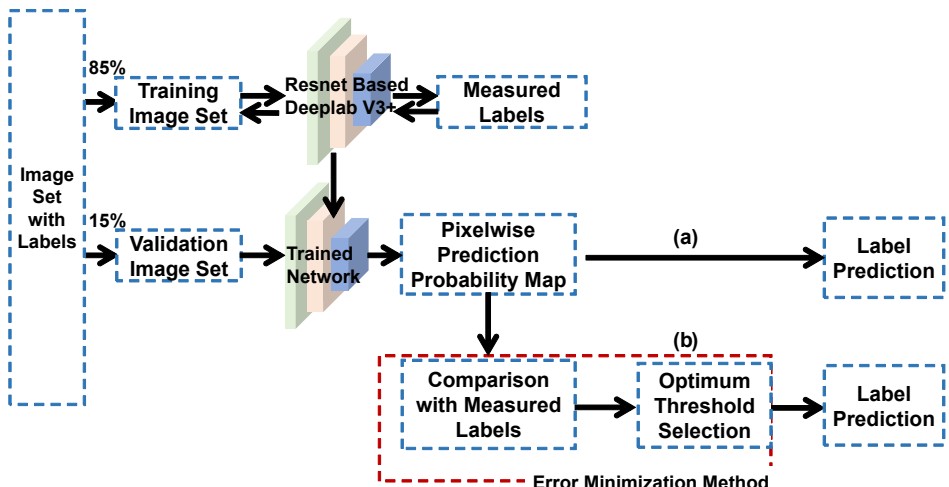

**Figure 4.** Diagram of the workflow we followed.

and set an initial learning rate of $10^{-3}$ that halves every 5 epochs. We found these rates provide better performance than using the same initial value but decreasing the rate by a factor of 10 every 10 epochs. Training process was repeated until it reached 30 epochs. The accuracy curve revealed no evidence of overfitting when 406 images (70 %) were used for the training set and 87 images (15 %) for the test set. Thus, to make the most use of the limited number of images, we used all 493 images (85%) for training the CNN for water prediction.

Figure 4 illustrates our general workflow to train and validate the CNN. As discussed in the next section, we introduced a new method to minimize false positives and improve CNN predictions.

## 3 Results

### 3.1 Raw CNN prediction for water

We evaluated the performance of the trained CNN by comparing the number of predicted water pixels to those "measured" as water in the validation set (also called "ground truth").

Following semantic segmentation, each pixel $(i, j)$ was assigned a probability $p_{ij}^k$ to belong to water ($k =$'w'), sky ($k =$'s') or background ($k =$'b'), with the dominant class being the one with highest probability. Thus, water pixels were described by the binary matrix:

$$w_{ij} = \Theta(p_{ij}^w - p_{ij}^s) \cdot \Theta(p_{ij}^w - p_{ij}^b), \tag{1}$$

where $\Theta$ is the Heaviside function ($\Theta(x) = 0$ for $x < 0$ and $\Theta(x) = 1$ otherwise). By definition, $w_{ij} = 1$ for water pixels and 0 otherwise.





The total number of predicted water pixels in a given picture was then

$$A_p = \sum_{i,j} w_{ij}. \tag{2}$$

Similarly, the number of measured water pixels was

$$A_m = \sum_{i,j} m_{ij} \tag{3}$$

where $m_{ij}$ is the binary matrix for water obtained from hand annotation and equals 1 for pixels identified as water and 0 otherwise.

  As shown in Fig. 5, raw CNN predictions ($A_p$) compared quite well to the measurements ($A_m$), in particular during flooding

conditions. The algorithm handled different lighting conditions well and clearly distinguished water and sky during storms. In general, there were no noticeable radiometric or geometric errors due to saturation, brightness, or hue caused by the sun or the GoPro curved lens introduced by different camera angles or distance to water.

### 3.1.1 Evaluation of accuracy

We quantified the accuracy of predictions for a single picture using three different metrics. The first one is the "accuracy ratio"

defined as the ratio of predicted to measured water pixels:

$$r = \frac{A_p}{A_m} \tag{4}$$

which just compares the size of the datasets without distinction between false positives or false negatives.

  The second one is the true positive rate or "sensitivity", defined as ratio of the number of true positives, given by the intersection $A_{m\cap p}$ of both datasets (i.e. measured water pixels also predicted as water by the CNN), and the number $A_m$ of

measured water pixels:

$$b = \frac{A_{m\cap p}}{A_m}, \tag{5}$$

where $A_{m\cap p} = \sum_{i,j} m_{ij} w_{ij}$.

  By definition, the sensitivity $b$ includes only true positives and therefore measures the effect of false negatives while neglecting false positives. In fact, the fraction of false positives is given by the difference:

$$r - b = \frac{A_p - A_{m\cap p}}{A_m}, \tag{6}$$

where $A_p - A_{m\cap p}$ is the number of false positive pixels.

  The third metric is the Intersection over Union (IoU):

$$u = \frac{A_{m\cap p}}{A_{m\cup p}} = \frac{A_{m\cap p}}{A_p + A_m - A_{m\cap p}} \tag{7}$$

which takes into account both sources of errors, false negatives and false positives.



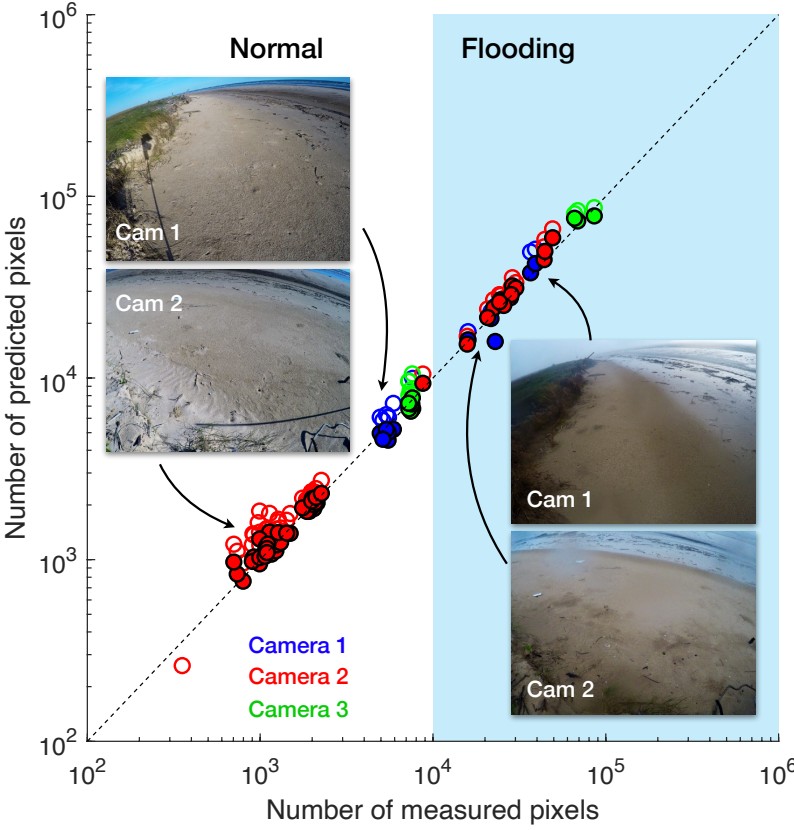

**Figure 5.** Comparison of CNN-predicted to measured water pixels for the 91 pictures in the validation set. Symbol's color represents the camera used. Both, raw CNN predictions (open symbols) and improved ones (filled symbols) are shown. Beach flooding seems to take place when the number of water pixels exceeds $10^4$.

Instead of analyzing the accuracy for individual images (it will be discussed afterwards), we use the Mean Absolute Percentage Error (MAPE) to provide a scale-independent metric of the global CNN performance.

The MAPE is defined as the mean over all images of the absolute deviations of $r$, $b$ and $u$ from the ideal value 1:

$$\delta r = \frac{100}{N} \sum_{n=1}^{N} |1 - r_n| \tag{8}$$

$$\delta b = \frac{100}{N} \sum_{n=1}^{N} |1 - b_n| \tag{9}$$

$$\delta u = \frac{100}{N} \sum_{n=1}^{N} |1 - u_n| \tag{10}$$





where $N$ is the total number of images and the subscript $n$ denote values for a single image. Therefore, the MAPEs quantify the deviations of the different accuracy metrics from the ideal, such that a perfect prediction would have MAPEs equal to 0. For simplicity, in what follows we will refer to the MAPE simply as the error of the respective accuracy metric.

We found the general error of the CNN, given by $\delta r$ and $\delta u$, was about 20%. The CNN was much better at predicting all measured water pixels, with only about 2% error in the fraction of true positives, as given by $\delta b$, which suggests most of the CNN uncertainty comes from false positives. Indeed, the average fraction of false positives $\overline{(r-b)}$ is about 22% as some regions have been mislabeled as water instead of background, for example, some frosts, wooden pieces, or wet sand areas nearby water (Fig. 6). One likely explanation is that the algorithm is trained in a manner that it tries to avoid false negatives while still allowing for false positives.

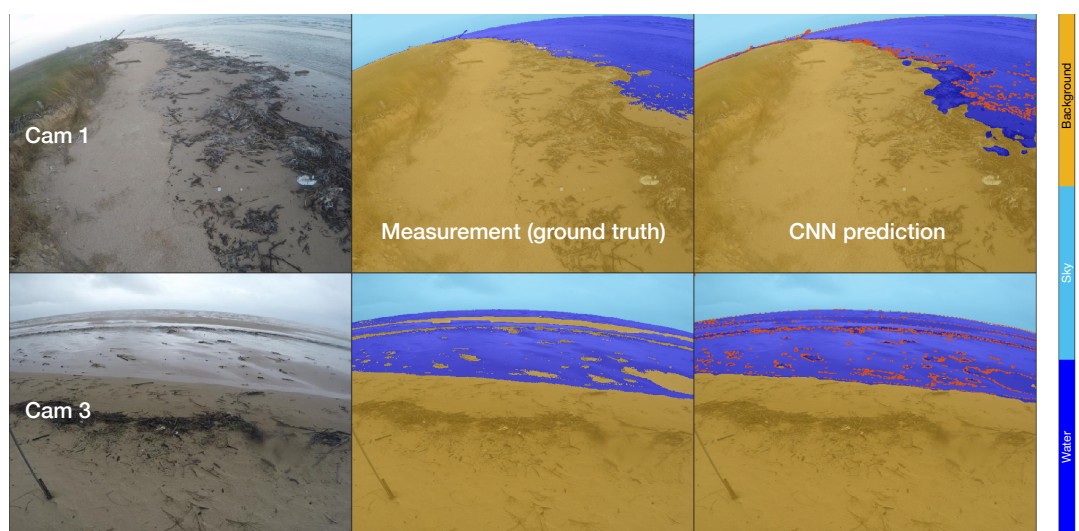

**Figure 6.** Examples of image segmentation during flooding conditions for cameras 1 (top) and 3 (bottom), showing the occurrence of false positives when the CNN mislabels wet sand as water. The red line superimposed on the CNN prediction represents the actual extension of water measured manually (center panels).

## 3.2 Improved CNN predictions

Our results demonstrate the utility of using a simple post-processing method to minimize the error $\delta r$ of the accuracy ratio $r$, by using only the classification probability. Although we focused on water segmentation, the proposed method could be extended further to the prediction of other classes as well. The central idea of such a method is to impose a threshold $p_t$ on the predicted water label $w_{ij}$ (Eq. 1), such that a given pixel $(i,j)$ is classified as water when the probability $p_{ij}^w$ is both the highest among the different classes and above $p_t$. The filtered binary matrix for water pixels in this case was given by

$$w_{ij}(p_t) = \Theta(p_{ij}^w - p_{ij}^s) \cdot \Theta(p_{ij}^w - p_{ij}^b) \cdot \Theta(p_{ij}^w - p_t). \tag{11}$$



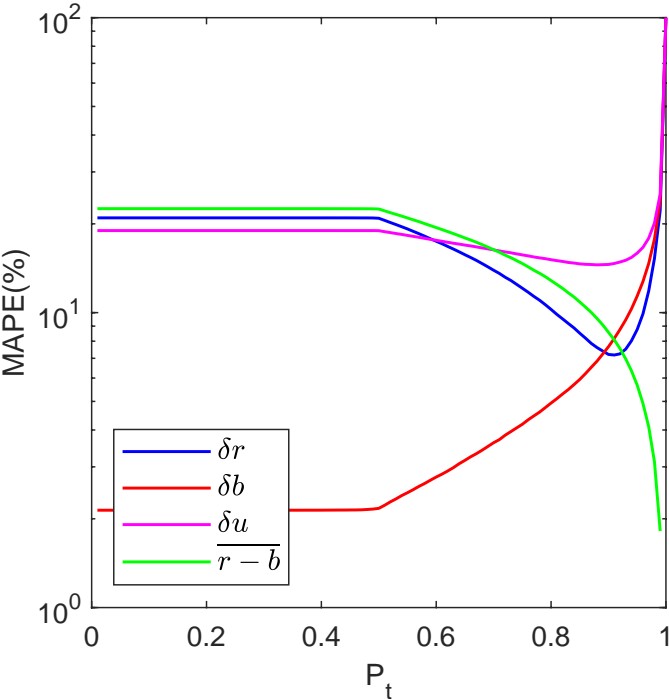

**Figure 7.** Mean absolute percentage errors (MAPE) of the accuracy ratio ($\delta r$), the sensitivity ($\delta b$) and the intersection-over-union ($\delta u$) as function of the threshold probability $p_t$. The average fraction of false positives $\overline{r-b}$ is also shown (in percentage).

From this definition it follows that the number $A_p$ of predicted water pixels and the ratios $r$, $b$ and $u$, as well as their corresponding MAPEs, (Eqs. 2,4-10) are all function of the threshold probability $p_t$. Imposing a threshold thus corresponds to a filtering of the raw CNN prediction. A zero threshold ($p_t = 0$) gives back the unfiltered values introduced in the previous

section.

For threshold probabilities below 0.5, we find the error of the accuracy ratio ($\delta r$) remains constant around 20%, which essentially corresponded to the unfiltered results (Fig. 7). For larger thresholds, $\delta r$ decreased and reached a minimum of about 7% for $p_t \approx 0.9$. Similarly, the error of the IoU ($\delta u$) reaches a minimum of about 15% also for $p_t \approx 0.9$. In contrast, the error of the sensitivity ($\delta b$) consistently increased with $p_t$ as more true positives are filtered out (Fig. 7).

The improvement in the accuracy ratio for increasing thresholds, as evidenced by a decrease in the error $\delta r$, was mainly due to the consistent reduction of false positives, shown by the average fraction of false positives $\overline{r-b}$ in Fig. 7. For water thresholds up to 0.9, the reduction of false positives outweighed the decrease in the sensitivity (i.e. higher error $\delta b$) led by the decrease of true positives (Fig. 7). However, for thresholds above 0.9, the loss of true positives became dominant and the CNN accuracy worsened as $\delta r$ increased.

We selected $p_t = 0.9$ as the optimum threshold probability for water segmentation in our validation set (Fig. 4), as it provided the most accurate results in terms of both the accuracy ratio and IoU (Fig. 7). Indeed, filtering the CNN results using $p_t = 0.9$




noticeably increased the accuracy of the CNN predictions and improved the accuracy ratio across the whole range of water conditions (Fig. 8).

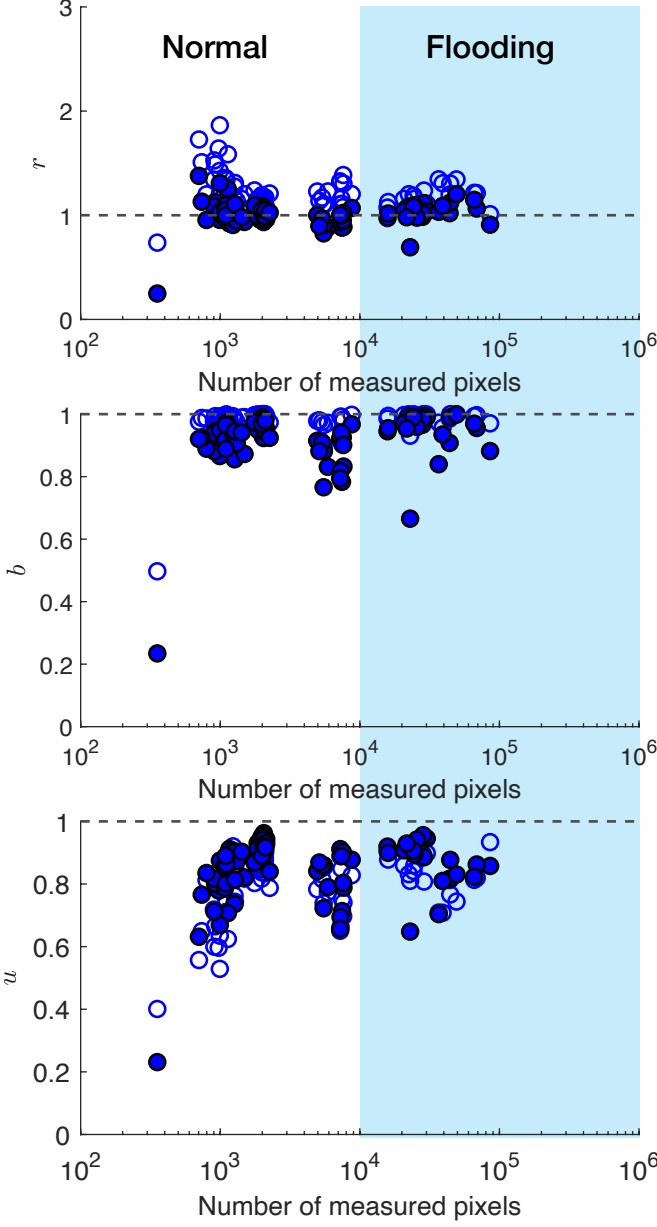

**Figure 8.** Accuracy ratio ($r$), sensitivity ($b$) and intersection over union ($u$) of the CNN predictions for water segmentation of individual images of the validation set as function of the number of measured water pixels. Both, raw CNN predictions ($p_t = 0$, open symbols) and filtered ones ($p_t = 0.9$, filled symbols) are shown.



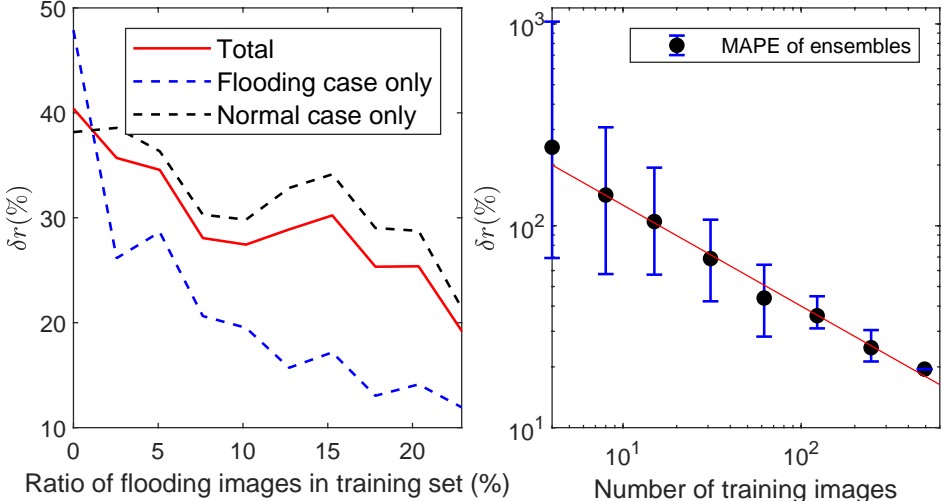

**Figure 9.** The Mean Absolute Percentage Error ($\delta r$) of the accuracy ratio of CNNs trained with different fractions of flooding images (left) and different number of images (right).

## 3.3 Effects of diversity and number of training images

Given how time consuming the training step can be, we were also interested in understanding how the accuracy of the trained CNN depended on two main factors, the diversity of images in the training set and the number of images in the training set.

Since we were primarily interested in identifying water, we divided the training set into two groups depending on the number of water pixels: "flooding images" with more than $1.5 \times 10^4$ water pixels and "normal images" otherwise (see pictures in Fig. 5). The number of training images was kept constant to 393 for every trained CNN, while the fraction of flooding images

was increased from 0% to about 23%. Each trained CNN was then tested with the validation set, which has 21 flooding images and 70 normal images.

As expected, the accuracy of the predictions for flooding images in the validation set increased with the proportion of flooding images in the training set, with the error $\delta r$ quickly decreasing from about 50% for no flooding images to about 12% for 23% of flooding images (Fig. 9). Surprisingly, the accuracy of the predictions for normal images in the validation set also

improved by almost 50% (with $\delta r$ decreasing from 40% to about 20%), despite the decrease of the fraction of normal images in the training set, which led to a significant improvement in the general accuracy of the CNN (Fig. 9).

Furthermore, to investigate the effect of the number of training images on CNN accuracy, we trained it using a random subset of $N$ different images from the total of 493 images available for training, roughly doubling N from 4 to 493 (Fig. 9, right). For each value $N$, we repeated the training $M$ times using different random samples to randomize the image type and account for

changes in the fraction of flooding images. We chose $M = 50$ for $N < 100$, decreasing $M$ for larger training sets ($M = 10$ and 5 for $N = 123$ and 247, respectively) to save training time. Each of the trained CNN was then applied to the whole validation set to evaluate the accuracy.



Again as expected, the accuracy of the predictions increased with the size of the training set (Fig. 9), with a large error $\delta r$, in the range 100-1000%, when using only 4 training images down to about 20% for the total 493. We found that $\delta r(N)$ decreases

with $N^{-1/2}$ which points to the random nature of the convergence of the CNN prediction.

## 4  Discussions and Conclusions

Our results demonstrate that CNN-based image segmentation is a viable method to identify water in complex coastal imagery. We found the mean absolute percentage error (MAPE) of the accuracy ratio (i.e. the ratio of predicted to measured water pixels) to be about 20%. This error can be further reduced to about 7% after applying a novel method for filtering false positives during

post-processing. After filtering, we found the MAPE of the true positive rate (sensitivity) to be about 8%, a value similar to the average ratio of false positives. We also found the accuracy of the filtered CNN predictions to be relatively independent of the amount of water in the images and it tended to be better for images with a larger fraction of water, corresponding to flooding conditions (Fig. 8). As for the lingering problem of false negatives, it may be better solved by interpolating/extrapolating a line across areas with greatest contrast, thereby connecting true-positive regions.

As expected, the CNN accuracy depended on the training data set. The accuracy increased with the number $N$ of training images, with the error decreasing as $1/\sqrt{N}$. In general, more than 100 images were needed to reduce the error below 50%.

Furthermore, the CNN performed much better when we increased the diversity of image types during training. This suggested that there is value in using additional metrics for quantifying image diversity, such as the multiscale structural similarity index measure (MS-SSIM) which evaluates the similarity between two images. For instance, we can ensure the diversity in the

training set by adding pair of images with very low MS-SSIM values.

As these methods become more fine-tuned, we expect that they will help improve predictions of coastal flooding and overtopping and thus enhance coastal resilience. Our work demonstrates the usefulness of in-situ camera systems with automatic image segmentation for real-time flooding detection. Furthermore, our findings elucidate the broad need for high-temporal resolution data to better understand the stochastic properties of flooding events and validate and/or improve widely used empirical

runup formulas. Indeed, the stochastic analysis and the comparison with empirical predictions is the focus of the second part of this study (Kang et al., in review).

*Author contributions.* O.D.V. designed the study. O.D.V., R.A.F. and T.H. installed the field equipment and carried out the observations. B.K. performed the CNN-based image segmentation. B.K and O.D.V. performed the analysis. B.K and O.D.V. prepared the manuscript with contributions from all co-authors.

*Competing interests.* The authors declare that they have no conflict of interest.



*Acknowledgements.* O.D.V. and B.K were supported by the Texas A&M Engineering Experiment Station.



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
