# Peer review of "Stochastic properties of coastal flooding events – Part 1: CNN-based semantic segmentation for water detection"

_EGUsphere, 2023_

## Author Response (AR1)

**Reviewer 1**

The manuscript "Stochastic properties of coastal flooding events – Part 1: CNN-based semantic segmentation for water detection" describes the development and accuracy of a Convolutional Neural Network (CNN) model that automatically detects water on the beach. The authors describe the training methodology and develop a post-processing method based on the classification probability (e.g., water, background, or sky) that improves the accuracy of the model's predictions. I find the paper to be well-written and concise.

This is a two-part submission, where Part 2 (also submitted to ESurf) uses the output from the CNN model developed here to evaluate flooding probabilities. I can see why the authors split the paper into two, as I believe aspects of the methodology are novel and there is a detailed validation process that would make one paper very long. However, I believe for this manuscript to be a standalone paper, the authors should contextualize the novelty of the developed algorithm (e.g., is this more automated and potentially faster than other classifying/ML models? Is the post-processing method unique?) and highlight the main contribution of the work to the field. Following on, it is hard to tell if this system was developed solely for validating the probabilistic analysis of Rinaldo et al., 2021 and what is described in Part 2 (Kang et al., in review). If that is the case, perhaps this is better as one paper. However, if the authors plan to use this technique in future applications, a discussion of the applicability/strengths to other sites could also help this manuscript stand alone. Some specific suggestions are found below.

Reply) Thank you for your constructive comments. Indeed, we find that using CNN-based image segmentation is faster than other methods for such a large number of images (>50,000) and don't require calibration or feature-extraction pre-processing. In total, we manually labelled only a 10% of the images (using a MATLAB function and a few days of work), and the trained algorithm was able to predict the other 90% in about two hours. Also, our error-minimization method is unique. Our main objective was to test the suitability of CNN-based image segmentation to handle complex coastal imagery and accurately identify water pixels in arbitrary locations, with the goal of expanding our ability to detect and predict water overtopping. Of course, this work stands alone and it is not tied to the validation of Rinaldo et al., 2021, which is exactly the reason of submitting a two-parts paper.

**General Comments:**

The introduction section is missing material that contextualizes the broad use and long history of using optical remote sensing for coastal-based field work. The way that Lines 27 – 35 reads suggests that camera-based systems are new tools for determining wave runup/shoreline position. Remote sensing wave runup using cameras (i.e., optical remote sensing) has been around for over 20 years now (see Holman and Stanley, 2007) and the authors should at least mention these in their introduction to camera systems for context. Time stacks of wave runup from cameras (e.g., how the Stockdon et al., 2006 R2% wave parameterization was generated) have long been manually digitized or QA/QCed, although there are also detection methods using edge detection algorithms, intensity thresholds, and other filters (https://github.com/Coastal-Imaging-Research-Network/CIRN-Quantitative-Coastal-Imaging-Toolbox). Recently, the US Geological Survey has indeed also moved towards leveraging the strengths of machine learning (Palmsten et al., 2020). I think this information can strengthen the authors' position that automated methods for classification are important.

Reply) We acknowledge the previous use of optical remote sensing in coastal-based fieldwork and the importance of providing a comprehensive historical context in our new modified introduction, where we added the following:

(...) Specially designed camera systems are now commonly used to monitor local wave runup, aiding in the refinement of empirical formulas and the increased understanding of runup's stochastic properties.

In nearly four decades, the evolution of optical remote sensing technologies has revolutionized wave runup monitoring, moving from manual digitization (Holman and Guza, 1984) to modern camera-based systems that efficiently capture wave runup and shoreline (Holman and Stanley, 2007).

(...) More recently, the US Geological Survey has also begun to harness the power of SVM to augment the edge detection algorithm to identify the runup edge (Palmsten et al., 2020).

Following on, other types of camera systems use geometric corrections and ground control points to measure elevations/water levels/wave runup. Can the authors explain and contextualize the strengths of a system that is only measuring the percent of beach flooded? Is this tool intended to be for long-term monitoring/deployment? Are the authors planning on using it for beach-flood validation in other locations?

Reply) Our goal in this work was to test the method for water detection in complex high-resolution imagery (arbitrary light and weather conditions) and not necessarily to properly estimate flooded area. We found that approximating the water area fraction based on the water-pixel fraction is enough for a statistical analysis of beach overtopping events, although of course it does not provide actual water area or elevation. This work can be easily expanded and improved in the future by combining it with hydrostatic pressure sensors at the beach and adding photogrammetry methods.

Terminology: The authors motivate their work discussing "flooding due to wave runup" and understanding the properties of "runup-driven wave events." I just want to make the point that the water levels the authors capture do indeed have runup in them, but they're total water levels (runup + still water level) as the authors are not removing the still water level signal, so the flooding is generated by other components related to the still water level (e.g., storm surge, sea level anomaly) not just the wave runup.

Reply) We fully agree other factors also contribute to flooding. We have removed these ambiguous terms from the manuscript.

Is the intent of this system to only look at low level events? I find the use of the terminology "coastal flooding" (in the title) a little broad and misleading, as it seems this is really looking at "beach flooding."

Reply) The method is general and can thus be applied to arbitrary large overtopping (flooding) events, which in our opinion justifies the title. It just happens that most of the events we captured on this location consisted of beach flooding. However, these apparently minor beach flooding events can lead to broader coastal flooding in lower-elevation areas.

Figure 9 displays a decreasing trend in the mean absolute percentage of error for different flooding % images and the number of training images. If the accuracy keeps decreasing, why not see where it "bottoms out"/becomes stable, i.e., where an increase in the number of training images or ratio of flooded images in training set doesn't affect the MAPE anymore? Maybe the authors also want to include a plot here that shows the increase in computation time relative to variation in # training images to show why a certain value is chosen, rather than the maximum decrease in MAPE.

Reply) We extended Figure 9 to include more data points (up to the maximum number of training images ~500), aiming to illustrate where the MAPE reaches stability. Furthermore, it only took about three hours to train and run the CNN model with the maximum number of training images we had (500 images) so we didn't evaluate the computation time as function of the size of the training set.

How computationally efficient is this entire effort?

Reply) Roughly three hours to train and run the CNN model for 51,000 images using an external GPU (NVIDIA GTX 1660TI with 6 GB GDDR6 memory).

Specific Lines:

Line 17 - 19: I am confused by this citation, as the Kang et al. paper's (Part 2 of this work) focus is not this.

Reply) The referee is correct. We corrected it in the revised version.

Figures 5 and 8 – open circles with closed circles plotted on top of them make the open circles hard to see.

Reply) We adjusted these figures.

Label Fig 8 and 9 with a,b,c; a,b, respectively instead of using "left" or "right."

Reply) Done.

I didn't see any data availability statement or a location where codes could be found – please see the journal's Data Policy.

Reply) Data is available in the Texas Data Repository (TDR), doi.org/10.18738/T8/TJPQA0.

**Reviewer 2**

**General comments**

The authors present an application of a Convolutional Neural Network to the segmentation of water from coastal imagery. The performance of the model and its sensitivity are quite thoroughly explored.

The general approach and description of the CNN, its training and validation are fairly clear and easy to follow. Something I had expected but is not mentioned in the current manuscript (or part 2) is the mapping of pixels from the image to real-world coordinates. Given the eventual purpose of the segmentation, mapping to real-world coordinates seems like a very useful thing to do as it would result in the actual height the waves have reached instead of just identifying which pixels contain water. And if it is not possible or feasible, it would be interesting to the reader to explain why.

Reply) We thank the reviewer for her helpful comments and suggestions. Our goal in this work was to test the method for water detection in complex high-resolution imagery (arbitrary light and weather conditions) and not necessarily to properly estimate flooded area. We found that approximating the water area fraction based on the water-pixel fraction is enough for a statistical analysis of beach overtopping events, although of course it does not provide actual water area or elevation. We agree that our work can be easily expanded and improved in the future by combining it with hydrostatic pressure sensors at the beach and adding photogrammetry.

In the manuscript, the term flooding is used a lot in a confusing way, at least for this reviewer. The introduction mentions flooding causing damage to coastal communities and infrastructure, which I then interpret as flooding of the hinterland (say landwards of the beach or dune crest). Going on the camera views shown in Fig. 2, the camera view contains the beach up to the start of vegetation, so from the images one can detect whether the beach itself is submerged (flooded). Furthermore the manuscript also mentions 'flooding events driven by wave runup', which is confusing from a semantics standpoint, as strictly speaking as soon as it leads to flooding (of the hinterland), it is by definition wave overtopping instead of wave run-up. In short, the authors should make more explicit and consistent what they mean with the term flooding, and what is of interest considering the context, namely high-frequency nuisance flooding that leads to damage to infrastructure and coastal communities.

Reply) We agree with the referee. We removed ambiguous expressions relating flooding and runup and extended our first introductory paragraph as follows:

Coastal flooding can cause significant damage to coastal infrastructure, communities, and saltintolerant ecosystems. By definition, flooding occurs when extreme water levels—due to a combination of high tide, wave runup and/or storm surge—exceed a natural or artificial threshold, e.g. a beach berm, dune, seawall. The frequency and severity of coastal flooding is expected to increase with the acceleration of sea level rise (Nicholls et al., 2011; Vitousek et al., 2017). In order to respond to and minimize the damage from coastal flooding, it is crucial to determine the frequency and intensity of flooding events at different locations and identify the physical factors behind them (Hallegatte et al., 2013; Moore and Obradovich, 2020). This is particularly relevant for highfrequency and low-intensity nuisance flooding not directly associated to large storms and thus difficult to predict and detect (Moftakhari et al., 2018).

**Specific comments**

1. L011: "Coastal flooding can cause significant damage to coastal ecosystems..." depending on your viewpoint, one could argue that coastal flooding is – or should be – a natural part of that

very ecosystem (especially in the case of barrier island systems). So I do not think this is a very strong argument.

Reply) Here we are referring to the harm caused by more frequent coastal flooding on the salt-intolerant vegetation essential to the coastal dune and back-dune ecosystem. We clarified this point in the introduction (see text above).

2. L029: "to monitor coastal flooding" – I think it's more accurate to state that you are monitoring local wave runup (or actually total water level). The coastal flooding itself is not monitored with this setup, right?

Reply) Yes, we agree. We corrected it in the revised version.

3. L090: "...weights in the cost of accuracy." should be "...weights at the cost of accuracy."

**Reply) Done.**

4. L115: 'images' should be 'image'

**Reply) Done.**

5. Fig 6: I don't see a superimposed red line in the center panels mentioned in the caption. (only red pixels in the right panels).

**Reply) We corrected the capture of Fig 6 accordingly.**

L236: The automatic segmentation of water in coastal imagery itself will not directly improve prediction of coastal flooding/overtopping, just the monitoring of wave run-up in a specific location. Indirectly, that data can be used in a way that does improve prediction as well, but this does not directly follow from (part 1 of) the manuscript.

**Reply) We agree. We changed it to the following text:**

(...) This method can greatly enhance the monitoring of local wave runup, in particular after combining it with photogrammetry and other ways to measure actual spatial data, thus contributing to improve predictions of nuisance coastal flooding and potentially enhancing coastal resilience. (...)

---

## Author Response (AR2)

I am pleased with the authors responses to my original review. I have a few remaining questions/corrections, which range from technical to minor.

Relating to the extended analysis of the MAPE of the accuracy ratio of CNNS trained with different fractions of flooding (Figure 9A): Did the authors test the MAPE accuracy with a proportion of more than 26% of flooding images?

Reply) For simplicity we kept the number of images constant and 26% was the maximum fraction of flooding images under that constraint.

There is also a decline in accuracy between 10 and 15%, so I'm wondering if the decline in accuracy the authors note continues past 26% or not?

Reply) We are not sure, however the decline in accuracy from 23% to 26% is much more noticeable and might indicate to an over-reliance of flooding images in the training, to the detriment to the correct identification of water fraction in non-flooding, or "normal" images. We look forward to testing this hypothesis in the future.

Following on, I believe that the authors see the lowest MAPE when 23% of the training set was composed of flooding images. It looks like the validation set was 21 flooding image and 70 normal images (line 215) making the percentage of flooded images in the validation set also 23%. Is this a coincidence? I suggest testing a different proportion of flooding images to ensure the best accuracy ratio proportion for training is independent of the percentage of flooded images of validation data used for validating/testing.

Reply) We indeed think it is a coincidence, however we can test a different proportion in future work.

What are the limitations of this approach? For example, if a camera moves, would the accuracy of the predictions be at risk?

Reply) The approach is very general. Note that we used three different cameras, with drastically different fields of view, and some of them indeed moved during the field campaign. However, the final result is quite homogeneous. See, for example, the lack of trend between the different symbol's color in Fig.5.

Line 240: Maybe I'm misunderstanding something, but isn't this statement contradictory? Accuracy of CNN predictions independent of amount of water in images, yet it was better for images with a larger fraction of water? This would tell me it might be dependent, because more water = more fraction of water?

Reply) These statements refer to two different things. The accuracy of the raw CNN predictions indeed is better for flooding images (Figs. 5 and 8A, open symbols; and Fig. 9). However, the accuracy of the filtered CNN predictions, depends little on the amount of water (Fig.5 and Fig. 8 top, filled symbols). We clarified this distinction in the revised version.

Line 252: is precise the right word to use here? Considering its just pixels from the images, I'm not sure it would be considered a precise observation of beach or back-beach overtopping. How is overtopping defined based on the imagery?

Reply) We removed the word precise to avoid confusion. By definition, the flooding of the beach or back-beach, as measured by the identification of water pixels on the beach/back-beach region, represent an overtopping event.

Line 253: I suggest editing this sentence to say, "This method has the potential to enhance...." Since it hasn't been combined with photogrammetry this is untested, so I would leave it as it has the potential, rather than it can enhance...

Reply) Done.

Figure 2: I suggest the authors label the figure to show which photos are considered flooding and which are considered normal.

**Reply) Done.**

Figure 3: I suggest the authors label "top row" and "bottom row" rather than just "top" and "bottom". Reply) Done.